# Evidence for Prepulse Inhibition of Visually Evoked Motor Response in *Drosophila melanogaster*

**DOI:** 10.3390/biology12040635

**Published:** 2023-04-21

**Authors:** Helgi B. Schiöth, Laura Donzelli, Nicklas Arvidsson, Michael J. Williams, Thiago C. Moulin

**Affiliations:** 1Department of Surgical Sciences, Division of Functional Pharmacology and Neuroscience, Uppsala University, 751 24 Uppsala, Sweden; 2Department of Experimental Medical Science, Faculty of Medicine, Lund University, 221 84 Lund, Sweden

**Keywords:** prepulse inhibition (PPI), *Drosophila melanogaster*, escape response, startle, anxiety, schizophrenia, invertebrates, sensorimotor gating, dizocilpine

## Abstract

**Simple Summary:**

In our study, we looked at a behavior called prepulse inhibition (PPI) in *Drosophila melanogaster*, commonly known as the fruit fly. For many animals, the sudden presentation of strong sensorial stimuli can induce a defensive response or motor reflex. PPI is a phenomenon where a small stimulus, named “prepulse,” is presented shortly before a larger stimulus, so the larger stimulus induces a weaker response than it normally would. This behavior is seen in many different types of animals and is used to study conditions such as anxiety and schizophrenia. For this study, the chosen stimulus is the sudden presentation of one-second darkness, or lights-off, shown to evoke an immediate locomotion response in *Drosophila*. Our research found that PPI can also be seen in adult flies, which has not been reported before. Additionally, we confirmed our results by showing that a drug that affects an important brain component, called the NMDA receptor, can change PPI in flies. We suggest that studying this behavior in fruit flies could help us understand how it works in other animals, including humans.

**Abstract:**

Prepulse inhibition (PPI) is a widely investigated behavior to study the mechanisms of disorders such as anxiety, schizophrenia, and bipolar mania. PPI has been observed across various vertebrate and invertebrate species; however, it has not yet been reported in adult *Drosophila melanogaster*. In this study, we describe the first detection of PPI of visually evoked locomotor arousal in flies. To validate our findings, we demonstrate that PPI in *Drosophila* can be partially reverted by the N-methyl D-aspartate (NMDA) receptor antagonist MK-801, known for inducing sensorimotor gating deficits in rodent models. Additionally, we show that the visually evoked response can be inhibited by multiple stimuli presentation, which can also be affected by MK-801. Given the versatility of *Drosophila* as a model organism for genetic screening and analysis, our results suggest that high-throughput behavioral screenings of adult flies can become a valuable tool for investigating the mechanisms behind PPI.

## 1. Introduction

The movement response to potentially harmful stimuli, or escape response, is one of an organism’s most basic sensorimotor functions. This behavior, defined as a prompt motor reaction to unexpected stimuli, is a fundamental feature of an organism’s defense against surrounding threats. During the escape reflex, a series of motor actions are triggered. These are a set of rapid responses to sensory stimuli, usually sudden and relatively intense, that may indicate danger. As such, this is a fundamental mechanism that many species use as their primary defense against predation. In *Drosophila melanogaster*, this behavior is mediated by the giant fiber system, which can be prompted by visual, olfactory, or mechanical stimuli. The subsequent response is the execution of stereotyped movements resulting in a motor escape response [1,2,3].

Behavioral tests that assess responses to potentially harmful stimuli play a crucial role in animal models of neuroscientific research. Since the abnormal features of defensive reactions have been linked to a range of neurological impairments in humans, the intricacies of these behaviors are extensively investigated in translational studies [4,5]. They have been used to decipher the mechanisms of various disorders, most notably anxiety [6], schizophrenia and bipolar mania [7], Tourette Syndrome [8], auditory system deficits [9,10], and posttraumatic stress disorder [11]. Given their significance, these tests are fundamental for the study of brain disorders in many model organisms. Among the various paradigms exploring defensive reactions, our study focuses on two of the most explored in the literature: prepulse inhibition (PPI) and desensitization to the startle-inducing stimuli, a main component of habituation learning.

PPI occurs when a weak prestimulus, presented shortly before a stronger stimulus, can, to some extent, suppress the motor response typically evoked by the stronger stimulus. This relative response reduction is a measure of sensorimotor gating, reflecting the activation of innate regulatory central circuits [12]. In patients, altered PPI has been observed to be linked to a range of brain disorders, including schizophrenia [13], Huntington’s disease [14], obsessive compulsive disorder [15], Asperger’s syndrome [16], 22q11 syndrome [17], and Fragile X syndrome [18]. This association has been the subject of extensive investigation over the past two decades, encompassing both animal and human studies, as PPI seems to be highly conserved across species. In addition to humans [19], this behavior has been described in various organisms, including rodents [12], non-human primates [20], and even in the invertebrate model *Tritonia diomedea* [21]. One report describes acoustic PPI in *Drosophila* larvae [22]; however, to the best of our knowledge, no studies have been performed demonstrating this behavior in adult flies.

Moreover, the habituation of the escape response and movement reaction has been described for almost four decades [23], generating comprehensive theoretical models defining the steps of sensory-motor network processing [24,25]. It is defined as a non-associative learning mechanism in which the recurrent presentation of the arousing stimulus gradually induces the inhibition of its initial motor response. In *Drosophila*, the habituation of visually evoked giant fiber response is a well-studied behavior, and many of the anatomical features and physiological properties have been well documented [25]. The response assessment is traditionally recorded as action potentials in the fly’s leg extensor and wing depressor muscles, which rely on costly electrophysiological equipment [26]. To circumvent this limitation, the jumping–landing reaction, a stereotyped motor pattern of legs and wings, is also used as an outcome for habituation [27]. However, the existing methods for measuring this motor reflex depend on subjective evaluation by trained observers, task-limited video tracking systems, or complex sound-based equipment [27,28,29]. Thus, we aimed to employ an easy-to-use and reproducible method for the assessment of the escape response, which would also allow for more versatility.

Recently, our group developed a novel platform for high-throughput screening of the flies’ light-based behaviors, named DISCO (*Drosophila* Interactive System for Controlled Optical manipulations), which is able to record second-by-second movements from individual flies [30]. Making use of this device, we showed that one-second lights-off stimuli increased *Drosophila*’s locomotion for both red-eyed and white-eyed flies, of which populational response ratios were remarkably similar to that from visually evoked giant fiber escape responses [30]. Notably, we also showed that a fly model of Fragile X syndrome (*Fmr1* null allele), which displays deficits in sensory responses [31], does not exhibit increased motor response after lights-off stimuli [30].

In this study, we employ the same platform to demonstrate the first evidence of PPI in adult *Drosophila melanogaster*. First, we reproduce the previous findings, demonstrating that immediate motor response to the lights-off stimuli (i.e., within the initial 3 s) can be detected by the device. Then, we develop a protocol where the usual movement response is inhibited by a prepulse stimulus (i.e., by partially turning off the LEDs, thus exhibiting only a weaker dim light) before the complete lights-off stimulus. Additionally, we show that the reduction in the movement response induced by repeated stimuli presentation can be observed by using the same apparatus.

Notably, the N-methyl D-aspartate (NMDA) receptor antagonist MK-801 was able to influence *Drosophila* behavior in both PPI and stimuli desentization. MK-801 is commonly used in preclinical studies to model phenotypes of schizophrenia in animals, such as sensorimotor gating deficits [12] and, importantly, impaired prepulse inhibition [32]. Thus, our results demonstrate the potential use of *Drosophila* as a model organism for further studies on PPI behavior and its modulation in sensory and motor systems.

## 2. Materials and Methods

### 2.1. Fly Strains and Maintenance

Experiments were conducted using 5 to 7 days old CSORC-strain *Drosophila melanogaster* flies (originating from CantonS and OregonR-C fly lines, Bloomington Stock Center, Bloomington, IN, USA). When indicated in the text or figure legends, we also conducted experiments using 1 to 3 days old flies (here called “younger flies”), collected at least 24 h after eclosion, in comparison to the usual 5 to 7 days old flies (here called “older adult flies”). Unless stated otherwise, all experiments included both male and female animals. The flies were maintained in vials at 25 °C, 12:12 h light/dark cycle, 60% humidity, and fed with Fisherbrand Jazz-Mix *Drosophila* food supplemented with 8.3% yeast extract (Fisher Scientific, Lund, Sweden). Importantly, although previous studies have shown that white-eyed flies are more responsive to lights-off stimuli [30], red-eyed flies were used in this study to increase the applicability of the results to a broader range of researchers. Throughout the study, experiments were performed within the light phases (ZT 0–12) of the light/dark cycle, covering the full scope of manipulation and testing of the flies.

### 2.2. NMDA Receptor Antagonist

To induce deficits in sensorimotor gating, flies were treated with the NMDA antagonist MK-801/Dizocilpine (Sigma-Aldrich, Stockholm, Sweden). Prepulse inhibition experiments were performed by treating the flies with MK-801 (0.6, 0.3, or 0.15 mM) for 24 h via homogenization in the fly food, from a 5 mM stock solution, during the preparation of the instant food mix previously described. The chosen concentrations were based on a preceding study from our group [33], where MK-801 effectively replicated hyperlocomotion phenotypes similar to rodent models in a dose-dependent manner. For the stimuli desensitization protocol, we administered the highest concentration used in this study (0.6 mM), as higher dosages of MK-801 have been shown to have a greater effect on startle habituation in rodents [34]. Control groups were created by adding an equivalent amount of vehicle solution (distilled water) to the fly food for all experiments.

### 2.3. DISCO Platform

As previously described [30], the DISCO apparatus is based on the MB5 MultiBeam Activity Monitor (TriKinetics Inc., Waltham, MA, USA), which has 16 independent tube slots, each equipped with 17 infrared beams for movement assessment. Three LED lights are connected to the MB5 Monitor per fly slot, uniformly spaced, and regulated by an Arduino Uno microcontroller (Arduino, Milan, Italy), which in turn is programmed to communicate with the MATLAB environment version R2018b (MathWorks, Natick, MA, USA) through the Legacy NeoPixel Add-On Library. Figure 1A shows a summary of DISCO assembly.

To perform the experiments, flies underwent CO_2_ anaesthesia and were individually placed in glass tubes sealed with cotton buds and a lid. The tubes were horizontally positioned in the apparatus, which constantly recorded the location and movement of the flies. After a 30 min acclimatization period, the light stimulation started, following the protocols described in the Results section. After the completion of all stimulation trials, raw data from the activity monitor device were analyzed by custom MATLAB routines.

### 2.4. Movement Response, PPI and Stimuli Desensitization

The number of beam crossings (counts) was measured by the DISCO apparatus for each second, and the lights-off stimuli presentation was controlled and recorded by the MATLAB-Arduino system. The raw data output from DISCO was converted to CSV files using the DamFileScan software from TriKinetics. A custom-made MATLAB routine was used to calculate the movement response to stimuli by subtracting the average movement counts for 3 s after a given stimulus from the baseline measure 3 s before the same point, disregarding the exact second in which the stimulus was presented, here named as startle index. Similar MATLAB routines can be found as Appendix A in a previous article describing the DISCO platform [30].

The calculated index allows for the visualization of the increase (or decrease) in movement, taking into consideration the baseline movement just before the stimuli. In simpler terms, positive index values denote an increased motor response, negative index values indicate a diminished response, and null values signify no deviation from the baseline response. It is similar to the delta movement index described in the original DISCO article [30], but with a focus on the immediate (3 s) response after the lights-off stimulus.

The responses to the lights-off stimuli were estimated after constant white-light illumination for 30 min before presenting the stimulus, and then starting a new trial loop (Figure 1B). This procedure was repeated for eight trials, and the movement response for each one was calculated (Figure 1C and Figure 2). The number of trials was chosen to optimize experiment duration while considering that flies, prone to dehydration [35], lacked humidity access in the testing tubes. Moreover, in our previous report [30], we conducted 6 trials (3 h) of a similar protocol without issues, leading to the arbitrary selection of 8 trials (4 h) for this study. As no noticeable effects on fly activity or survival occurred, the protocol was maintained. The average of all trials was used as a control response baseline for startle-inducing stimuli in subsequent experiments.

For assessing PPI, we modified the startle-inducing trial protocol, so a dim light was presented before the full-darkness stimulus, generated by turning off one of the three LEDs that illuminate the flies (Figure 3A). As done previously, we measured the response index for eight consecutive trials, but now presenting the partial-light pulse 10, 5, or 3 s before the movement-inducing stimulus. To evaluate the effect of MK-801 on PPI, for each concentration and prepulse/stimuli interval, independent experiments with controls and treatment groups were performed (Appendix A) and then grouped together for statistical analysis (Figure 3B). A separate control experiment verified if MK-801-treated and untreated flies could be directly compared, as we found that flies receiving the highest concentration (0.6 mM) had a similar baseline response to untreated flies for the startle-inducing stimulus (Appendix A).

For the stimuli desensitization assay, we modified the stimulation algorithm to present a repeated 1 s lights-off stimulus intercalated with 1 s light-on stimuli for a minute (i.e., 1 min blinking lights at 1 Hz). This stimulation was repeated for nine trials, 10 min apart (Figure 4A). The movement response for each trial was calculated by subtracting the average movement counts for 3 s after the end of the entire stimulation trial from the 3 s of baseline movements.

### 2.5. Statistical Analyses

We performed D’Agostino and Pearson normality test analysis on our initial sample to verify if the calculated values startle index follows a normal distribution. As it passed the normality test, subsequent analyses were made using parametric tests. For experiments where we investigated if the movement response was above baseline (i.e., startle index = 0) (Figure 1C and Figure 2), a one-sample t-test was used to detect the difference from 0. A two-way ANOVA analysis was applied in experiments testing the overall effect of MK-801 exposure in relation to the controls considering all trials (Figure 2, Figure 3C and Figure 4B), followed by Holm–Sidak posthoc tests. In Figure 3B, to investigate the MK-801 concentrations effects on different prepulse intervals (PPI), we have performed one-way ANOVAs followed by Holm–Sidak posthoc tests within each interval assessment. This approach was chosen as the experiments between different PPIs are entirely independent, but comparisons within identical intervals should undergo multiple comparison corrections, as same-interval control groups were clumped together. Importantly, the Holm–Sidak method, akin to Bonferroni, corrects *p*-values for multiple comparisons while offering greater statistical power [36]. Although it does not compute adjusted confidence intervals for the mean differences, it was chosen since our focus was not on the effect’s magnitude but rather on confidently determining the presence of PPI and the possible impact of MK-801 exposure. Nevertheless, unadjusted 95% confidence intervals of the differences were calculated for consultation. The statistical tests used for each experiment and sample sizes are described in the legend of their respective figures. All analyses were performed using the GraphPad Prism 8 software. For a detailed examination of the statistical results, refer to the tables in the Appendix A.

## 3. Results

### 3.1. Automatic Detection of Escape Reflex

For the experiments, we used the DISCO apparatus, an infrared-based activity monitor combined with automated control of light stimuli [30]. Importantly, it can detect the movements of flies inside individual glass tubes, and the activity monitor chosen has 17 independent infrared beams, so the unit was able to record movements at any location within the length of each tube. Custom software was used to set specific illumination protocols using the LED light stripes coupled to the device (Figure 1A).

The protocol was designed to evoke movement responses by exploiting the escape reflex after the presentation of visual stimuli, in this case, transitory darkness perceived by the fly as the shadow of a nearby predator [29,37]. DISCO presented to the flies a constant white-light illumination for 30 min, which was turned off for one second before restarting a stimulation loop (Figure 1B). This procedure was repeated for eight trials, and the startle index for each trial was calculated (Figure 1C,D, see Section 2). We observed that the flies presented a movement increase in response to the sudden lights-off stimuli. One sample t-test analysis of the 8-trial responses average showed that it was significantly increased from baseline (Figure 1E; mean ± SEM 8-trials: 0.17 ± 0.02; *p* = 0.0004).

### 3.2. Characterisation of the Visually Evoked Response

Next, we sought to further characterize the nature of the immediate movement response to lights-off stimuli. As it has been shown that male and female flies can differ in many light-related behaviors, including daily activity and motion responses [38,39,40], we separated males and females from our sample (Figure 1D) and assessed if there were sex variations in the startle index (Figure 2A). Two-way ANOVA showed no significant difference among male and female responses (*p* = 0.985). There was a significant effect of the trial number on the startle index (*p* = 0.012), likely caused by the incidental reduction of movement response on trial number 3 from both groups; however, no effects of sex-trial interactions were observed (*p* = 0.467). Moreover, mating experience induces physiological changes that also can modulate the flies’ locomotion, especially for female *Drosophila* [41]; thus, we compared the reaction to lights-off stimuli from virgin and mated female flies (Figure 2B). Likewise, no significant differences could be found among the conditions (*p* = 0.809). With this independent sample, no trial number or interaction effects were detected (*p* = 0.751 and *p* = 0.361, respectively). Additionally, we tested if younger adult flies (1 to 3 days old) behaved differently from older adult flies (5 to 7 days old, same as Figure 1D) when the stimuli were presented, as the maturation of the nervous system could affect sensory processing (Figure 2C). Once more, the two-way ANOVA analysis did not find any significant differences between the two age groups (*p* = 0.373), and no influence from trials (*p* = 0.394) or age-trial interaction (*p* = 0.096). The mean values of startle index from the 8 trials for all groups were significantly greater than baseline response (*p* = 0.027 and *p* = 0.024 for virgin and mated females, respectively; for all other groups, *p* < 0.0001).

### 3.3. Prepulse Inhibition of the Movement Response

PPI refers to the reduction in motor response that occurs when a weak, prepulse stimulus is presented prior to the arousal-inducing one. Thus, to evaluate the ability of DISCO to detect PPI in adult *Drosophila melanogaster*, we modified our trial protocol to include a weak prestimulus in the form of a dim light (generated by turning off one of the three LED lamps to which each fly is exposed) presented before full-darkness (all lights-off) stimulus (Figure 3A). As done in the initial experiments (Figure 1C), we measured the movement response for eight consecutive trials, but now presenting the partial-light pulse 3, 5, and seconds before the startle-inducing stimulus. Additionally, three concentrations of the NMDA receptor blocker MK-801 (0.15, 0.3, and 0.6 mM) were administered to the flies prior to PPI testing. To evaluate the overall effects of prepulse interval and MK-801 exposure, we conducted nine independent control-paired experiments—one for each prepulse interval and drug concentration (Appendix A). The resultant average responses of each group were subsequently analyzed together (Figure 3B,C).

Figure 3B shows the main effect of MK-801 exposure by presenting the average responses of each group side-by-side. The longer interval (10 s) was unable to inhibit the escape response, as the movement response was comparable to baseline responses with no prepulse presentation (at 10 s interval: Baseline = 0.17 ± 0.02; Control = 0.15 ± 0.02; MK-801: 0.15 mM = 0.15 ± 0.03; 0.3 mM = 0.22 ± 0.03; 0.6 mM = 0.19 ± 0.03. Values are mean ± SEM). This result is expected as organisms are less likely to perceive the prepulse as a warning signal as the interval between the two stimuli increases, then failing to inhibit the movement response to startle-inducing stimuli.

Nevertheless, the prestimulus presentation at 5 s interval was sufficient to abolish the motor reaction in control flies and for the 0.15 mM and 0.6 mM MK-801 groups, demonstrating inhibition of the escape reflex. This inhibition of movement response was partially reverted by MK-801 at 0.3 mM concentration (at 5 s interval: Control = 0.00 ± 0.02; MK801: 0.15 mM = −0.02 ± 0.0; 0.3 mM = 0.07 ± 0.02; 0.6 mM = −0.01 ± 0.03). Similarly, the shorter interval of 3 s suppressed the startle response in wild-type flies; however, it was partially reverted by the 0.15 mM and 0.3 mM MK-801 concentrations (at the 3 s interval: Control = −0.02 ± 0.015; 0.15 mM = 0.06 ± 0.03; MK-801: 0.3 mM = 0.05 ± 0.02; 0.6 mM = −0.01 ± 0.02). One-way ANOVAs and posthoc analyses within experiments with the same prepulse interval showed a significant effect of the 0.15 and 0.3 mM concentrations when compared to controls (*p* = 0.007 and *p* = 0.023 for 0.15 and 0.3 mM, respectively, Holm–Sidak test). Considering all prepulse intervals and administered concentrations, the two-way ANOVA analysis indicated significant effects of treatment (*p*_MK-801_ = 0.006), particularly for 0.3 mM MK-801 exposure (*p* = 0.002, Holm–Sidak posthoc test).

Figure 3C better illustrates the response curve for different prepulse intervals. We can clearly observe the U-shaped effect of different prepulse intervals, confirmed by the two-way ANOVA analysis (*p*_trials_ < 0.0001). Moreover, despite the significant effects of MK-801 exposure, there were no statistical effects of interaction between prepulse interval and exposure (*p*_interaction_ = 0.263), indicating that MK-801 may have a consistent influence on the startle response across tested prepulse intervals. All ANOVA tables and posthoc statistics can be found in the Appendix A.

### 3.4. Inhibition of Visually Evoked Response by Multiple Stimuli Presentation

Next, DISCO was employed to generate a protocol in which the startle-inducing stimulus is presented multiple times to induce a reduction of the motor response, an important component of habituation learning. Unlike PPI, this paradigm is well described in *Drosophila melanogaster* [25]. To prompt the inhibition of the startle response, we modified the stimulation algorithm to present repeated 1 Hz lights-off stimuli for one minute, for nine consecutive trials, with a 10 min interval between trials (Figure 4A). For trials 1–4, control flies presented movement responses comparable to those from previous experiments, which became gradually reduced in the subsequent trials (trials 5–9). Between-treatment evaluation of the movement response made by two-way ANOVA analysis showed that administration of the NMDA receptor antagonist MK-801 induced a distinct movement response (*p*_MK-801_ = 0.043). Independent of the groups, however, movement response significantly reduced over time and eventually reached negligible values (*p*_time_ = 0.0003), indicating that the response diminished by multiple stimuli presentation. No significant treatment x time interaction was observed (*p*_interaction_ = 0.52).

## 4. Discussion

The stereotypic movements of the fly jumping–landing response to looming shadows or lights-off stimuli have been extensively studied [29]; however, more complex behavioral protocols based on this paradigm are still limited in *Drosophila*. In this study, we utilized the DISCO platform, a straightforward LED and infrared-based setup for light stimulation and motion detection, to record movement responses for lights-off presentation to flies. We demonstrated that infrared-based locomotion detection can record the fly’s innate escape response, elicited by a one-second lights-off presentation. Our results were in line with previous findings, which showed that visually evoked increases in locomotion could persist for up to 30 s, with a peak occurring approximately 2–3 s after the stimulus [30], which coincided with the recording time frame used in the current assessments. Our findings revealed no significant differences in the responses of male and female, virgin and mated females, or younger and older flies. Due to the small variation in the outcome across those various conditions, these results demonstrate that our assessment of movements after lights-off is a robust method for measuring visually evoked responses. Thus, based on this behavior, we sought to develop and implement straightforward protocols to study PPI and desensitization to the startle-inducing stimuli.

PPI refers to the process by which a subthreshold, or less intense, prestimulus is delivered prior to the presentation of a more intense arousal-inducing stimulus, resulting in a reduced behavioral response. This phenomenon has been studied in *Drosophila* larvae, where acoustic stimulation was utilized to evoke motor responses, which were then hindered by the presentation of a shorter-duration stimulation [22]. Of particular interest is the fact that the stimuli used in this report were representative of the natural sound of wasps, a predator of flies, thereby establishing an interesting biological parallel with stimuli such as looming shadows or light-off stimuli that also mimic an incoming predator. However, to the best of our knowledge, no studies have explored this behavior in fully developed adult flies.

With that in mind, we developed a protocol for prepulse presentation preceding the movement-evoking lights-off display. Three independent groups of flies were tested with intervals of 3, 5 and 10 s between the weaker and the stronger pulse to evaluate the effects of gradually increasing the distance between the two stimuli. It is worth noting that the DISCO’s minimum recording resolution of one second, and the subsequent selection of a 3 s window for measuring movement responses, imposed certain limitations on the design of this protocol. In most PPI studies in humans and rodents, the interval between the prepulse and main stimulus is typically much shorter, ranging from a few hundred milliseconds to no more than two seconds [42]. However, our choice was motivated by initial reports of PPI in rodents that employed longer interval periods and documented some degree of response inhibition up to 4 s after prepulse presentation [43,44]. It was also observed in *Tritonia Diomedea* complete inhibition of the motor response with intervals of up to 2.5 s and partial inhibition of up to 5 s between the two stimuli [21], further supporting our protocol design. Based on these results, we decided to proceed with our approach.

Following this method, the DISCO platform was capable of detecting the PPI phenomenon in adult *Drosophila.* Our results demonstrate that the partial presentation of the stimulus (i.e., a dim-light display) was able to significantly inhibit the subsequent movement response in the flies when presented 3 and 5 s prior to the lights-off stimulation. However, the longer interval of 10 s did not produce the same inhibition. This outcome is in line with PPI observed in other animals, as organisms are less likely to interpret the prepulse as a warning signal as the interval between the two stimuli lengthens, subsequently failing to inhibit the startle response to the main stimuli.

Next, we tested the effects of MK-801 exposure on PPI. This compound is a selective NMDA receptor antagonist widely used in preclinical research and has been previously employed disruption of PPI in rodent models [32]. Notably, the binding site of MK-801-like drugs has been shown as essential for PPI-inhibition effects [45]. We observed that the prepulse effect was partially disrupted by exposing the flies to MK-801. We employed three different concentrations based on previous experiments describing the effects of MK-801 on *Drosophila* behavior [33]. First, 0.15 mM MK-801 partially disrupted PPI within the 3 s interval testing, although we did not detect significant effects across all other intervals. The 0.3 mM group also exhibited a significant PPI reversion for the 3 s interval and was the only exposure that exhibited a significant effect when considering all prepulse intervals. Conversely, MK-801 at 0.6 mM did not influence PPI in any of our analyses. This pattern of effect magnitudes across tested concentrations may be a noteworthy finding, as it might suggest an inverted U-shaped dose–response curve between NMDA inhibition and PPI disruption in flies.

Importantly, *Drosophila* features conserved homologues of key mammalian genes encoding NMDA receptor subunits, specifically NR1 and NR2, which exhibit common genetic and structural attributes [46]. For instance, the NR1 glycine-binding and NR2 glutamate-binding domains display conserved amino acid sequences [47,48,49], and functional similarities in NMDA receptors concerning signaling and neuronal plasticity are well established [33,49,50,51]. Moreover, pharmacological parallels between MK-801 interactions with mammalian and invertebrate NMDA receptors have been identified, including blocking NMDA-dependent processes [51,52,53] and targeting the conserved asparagine residue in NR1 subunits [48,49,50,53]. Thus, the observed effects of MK-801 not only reinforce the validity of the results but also highlight the potential of using *Drosophila* as a model system to further investigate the neural mechanisms underlying PPI. Nevertheless, we should keep in mind that although the 0.6 mM concentration had no significant influence throughout experiments, it would be ideal to also observe a smaller concentration with no effects to serve as a valuable baseline for comparison. While our findings provide valuable insights into the role of NMDA receptors in prepulse inhibition, we recognize they also raise further questions on their mechanistic details and suggest future studies to include a wider range of MK-801 concentrations to establish a more refined dose–response relationship.

We also evaluated the efficacy of DISCO in measuring a gradual reduction of the startle response by multiple stimuli presentation. This process, inherent in habituation behavior, is a well-established paradigm in *Drosophila melanogaster* [25]. It is defined as a non-associative learning mechanism in which an initial response towards a stimulus gradually fades after its repeated presentation. A range of habituation protocols has been created encompassing various sensory modalities, such as visual (giant fiber escape reflex, landing response); chemical (proboscis extension, olfactory jump, odor-induced leg response, experience-dependent modification of courtship); electric (shock avoidance); and mechanical (leg resistance, cleaning reflex) [25]. Here, we modified the DISCO algorithm to perform multiple lights-off presentations so that the response to stimuli was reduced over time. Unlike the usual protocols for flies, in which a long series of flashes is presented once [19,20], we applied consecutive trials of multiple stimuli, as commonly done for rodent models [43]. After four trials, the startle index reached null levels, indicating that the movement response was inhibited. These results suggest that the repeated presentation of the lights-off stimuli was capable of a response reduction. This change may be derived from a habituation learning process; however, other criteria should be verified in future studies to confirm this interpretation [54]. Interestingly, MK-801 influenced the response amplitude for the initial trials but ultimately did not prevent stimuli desensitization.

Additionally, the successful observation of PPI after using dim light as a prepulse stimulus suggests that it may be worth considering future studies on the potential influence of environmental factors, such as weather or daylight conditions, on prepulse inhibition. For instance, fluctuating light conditions in natural settings could affect sensory processing, potentially impacting the responses to various threatening stimuli. Exploring these connections could provide valuable insights into how environmental light settings may contribute to the sensory and behavioral responses across species.

Due to its simplicity, our approach to the assessment of PPI and stimuli desensitization can also contribute to a broader use of behavioral tests exploring stimuli-evoked movement responses in *Drosophila*. So far, efficient measuring of motor reflexes requires specialized equipment, and alternatives would involve intricate electrophysiological recordings of muscular response [26] or analyses of video footage dependent on highly trained experimenters [29]. Our results show that the reaction to visual stimuli can be measured by a straightforward locomotion-based approach, reducing the requirements for costly equipment or extensive training. Moreover, with the possibility of testing multiple individual flies, similar methods to those described here can be easily employed for high-throughput experimental settings.

Combined with *Drosophila*’s versatility and powerful genetic toolbox, further development of these tests can significantly contribute to drug screenings or genomic studies related to psychiatric disorders [55,56]. Many risk genes are conserved in *Drosophila*, so assessing sensory-evoked responses would allow for identifying molecular pathways related to brain pathologies, which could be explored as therapeutic targets. In fact, PPI screenings led to the identification of numerous candidate genes that influence sensorimotor gating, but the functional interactions among these genes remain mostly elusive [57]. Additionally, as impairments in habituation learning are largely present in neurodevelopmental disorders, it has been proposed that this behavior is ideal for high-throughput genetic investigations of such conditions [24].

Thus, the use of *Drosophila* models, in conjunction with innovative apparatus such as ours, presents a highly promising avenue for uncovering the machinery behind brain disorders. By enabling efficient examination of extensive gene sets and molecular pathways [42], this approach holds great potential for advancing our understanding of these complex processes. A recent study has served as a pioneering demonstration of this concept, utilizing sound-based scoring of the jumping response to evaluate nearly 300 genes associated with intellectual disability [28]. Despite these efforts, reports of this nature are still limited in the literature. With the novel protocol presented here, which enables the use of flies as in vivo screening models for sensorimotor alterations combined with learning tasks, we aim to contribute to filling this gap and furthering the goal of understanding the mechanisms underlying pathologies in the nervous system.

## 5. Conclusions

Our findings illustrate the utility of straightforward movement-based recordings for the measurement of PPI and stimuli desensitization in adult flies, offering a novel and accessible approach for investigating these foundational behaviors that have a strong connection to several human brain disorders, such as schizophrenia, anxiety, bipolar mania, and PTSD, among others. Moreover, our results indicate that NMDA receptors may play similar roles in prepulse inhibition across species. We believe further investigations, potentially integrating genetic tools, could contribute to unravelling the neural mechanisms behind the PPI paradigm. With further refinement and combined with the *Drosophila* toolbox for manipulating specific genes and neuronal circuits, behavioral tests such as those presented here can significantly advance our understanding of psychiatric disorders by enabling drug screenings or genomic studies.

## Figures and Tables

**Figure 1 biology-12-00635-f001:**
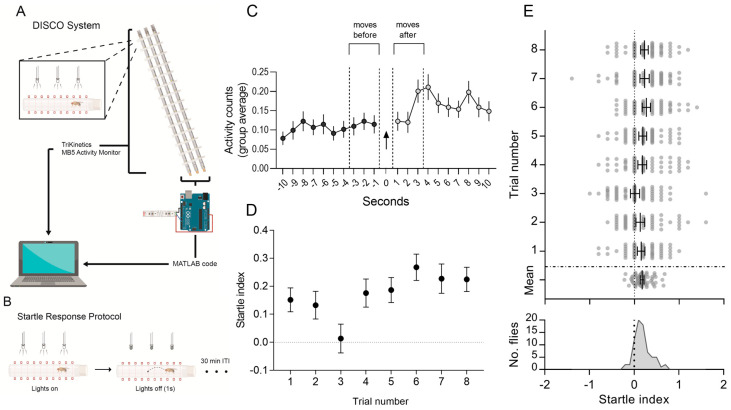
Automated detection of visually evoked motor response. (**A**). Schematics of the *Drosophila* Interactive System for Optical manipulations (DISCO). The system combines infrared movement detection and automatically controlled LEDs (figure adapted from [30]). (**B**). Each glass tube is aligned with three LEDs, which are turned off for 1 s to induce a visually evoked motor response, with a 30 min inter-trial interval (ITI) (**C**). The startle index was calculated considering the average movement counts of each fly three seconds before and after the lights-off stimuli presentation, represented by the arrow at t = 0 (figure adapted from [30]). (**D**). Flies were tested over eight trials consecutive trials, where group averages indicate an increased motor response for most stimuli presentation. Null startle indexes (dotted lines) denote measurements with no movement change. (**E**). Visualization of single fly responses across each trial indicated a highly variable individual behavior; however, the mean response for all trials (under the dashed line) shows a robust group effect (*p* = 0.0004, n = 74 male and female flies, one-sample t-test against 0). Histogram of the flies’ mean startle values show no significant departure from a normally distributed response (*p* = 0.052, D’Agostino and Pearson normality test). All central lines and bars represent mean ± SEM.

**Figure 2 biology-12-00635-f002:**
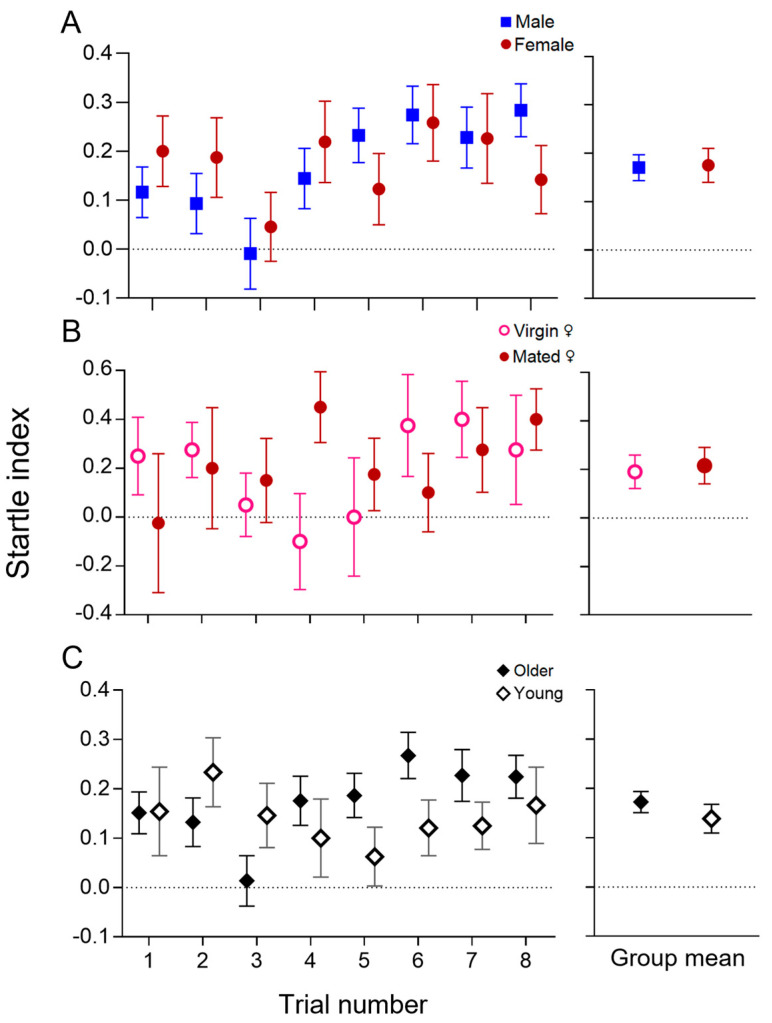
Motor response to lights-off stimuli for different conditions. (**A**). Adult flies tested previously (Figure 1D) were separated into male and female groups for assessment of possible sex differences. No significant distinctions among sexes were found (two-way ANOVA, *p*_sex_ = 0.985, *p*_trials_ = 0.012, *p*_interaction_ = 0.467, n = 43 males/31 females). Both sexes’ 8-trial averages showed a significant movement increase after stimuli (one-sample t-test against 0, *p* < 0.0001 for both groups). (**B**). Effects of mating were tested by collecting same-age virgin and mated females. Similarly, no differences in response were found (two-way ANOVA, *p*_mating_ = 0.809, *p*_trials_ = 0.751, *p*_interaction_ = 0.361, n = 8/group). Group averages also showed a significantly increased motor response (one-sample *t*-test, *p* = 0.027 and *p* = 0.024 for virgin and mated females, respectively) (**C**). Lastly, the movement responses from young flies (1–3 days old) were compared to those from adult flies (5–7 days old, same animals from Figure 1D). However, no significant differences were detected (two-way ANOVA, *p*_age_ = 0.373, *p*_trials_ = 0.394, *p*_interaction_ = 0.096, n = 74 older-adults/48 young-adults) and 8-trial means indicated that both groups responded greater than baseline (one-sample *t*-test, *p* < 0.0001 for both groups). Values are shown as mean ± SEM. Detailed statistical results can be found in the Appendix A.

**Figure 3 biology-12-00635-f003:**
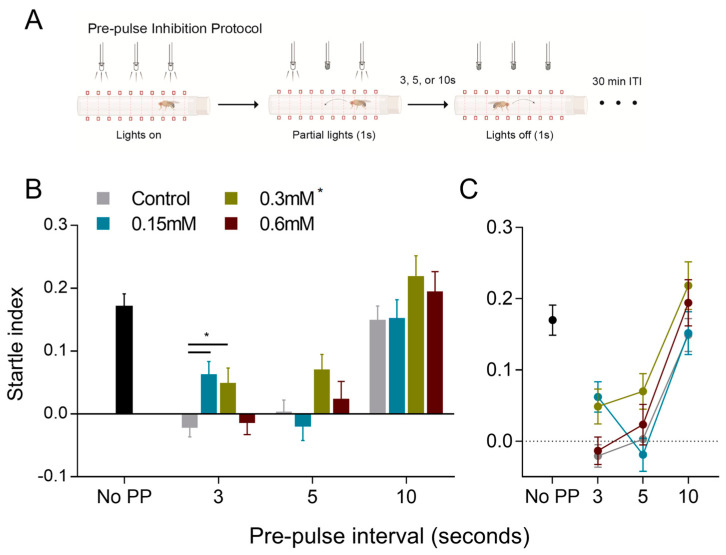
NMDA-dependent prepulse inhibition of motor response in flies. (**A**). The DISCO protocol was modified, so a dim-light prepulse was presented shortly before light-off stimuli. (**B**). Prepulse stimuli presented 3 s before were incapable of inhibiting the typical movement response. However, partial stimuli exhibition 3 and 5 s previous to the main stimuli were able to abolish the baseline response in controls, demonstrating prepulse inhibition (PPI). Considering all trials, MK-801 was able to have an effect on the movement of flies (average of 8 trials; two-way ANOVA, *p*_MK-801_ = 0.006; n = 38–81 flies/group), with clearer results at 0.3 mM concentration. Moreover, 0.15 mM and 0.3 mM MK-801 was able to significantly inhibit PPI within 3 s prepulse interval trials (*p* = 0.007 and *p* = 0.023 for 0.15 and 0.3 mM, respectively, one-way ANOVA with Holm–Sidak posthoc test). (**C**). As expected, the length of the prepulse interval modulated the startle response, as the longer 10 sec interval was unable to produce PPI (two-way ANOVA, *p*_trials_ < 0.0001). Moreover, no interaction effects between MK-801-exposure and were observed (two-way ANOVA, *p*_interaction_ = 0.263). Values are shown as mean ± SEM. * *p* < 0.05.

**Figure 4 biology-12-00635-f004:**
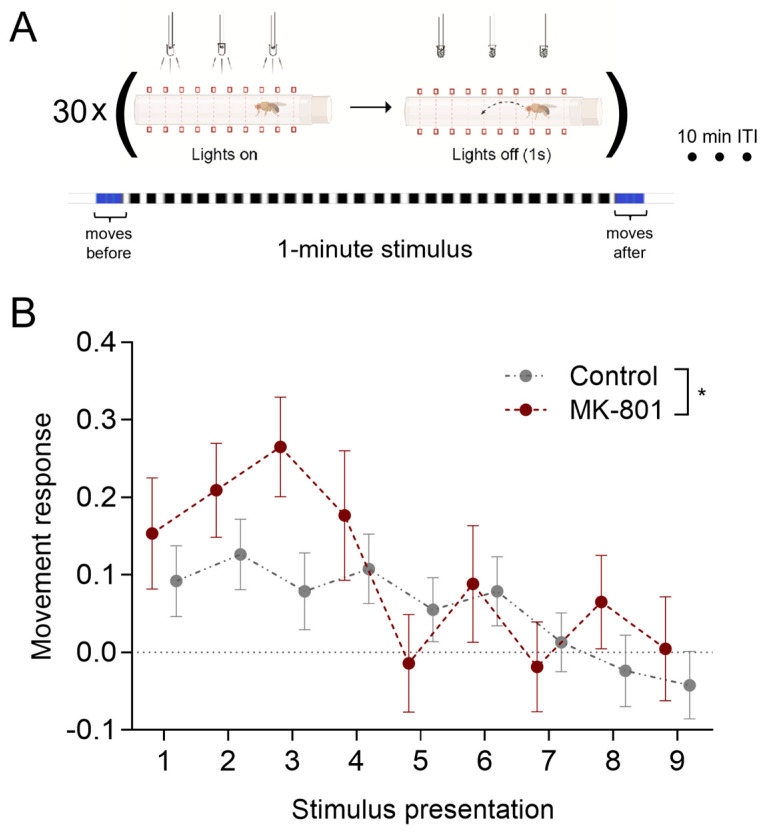
Desensitization to startle-inducing stimuli is affected by MK-801. (**A**). By repeatedly presenting the lights-off stimulus for one minute and by reducing the interval between trials of lights-off stimuli presentation, we tested if the motor response could be inhibited. (**B**). For the first four trials, both groups presented startle responses significantly over baseline and negligible responses afterwards, demonstrating the stimuli desensitization over time (two-way ANOVA, *p*_trial_ = 0.0006). Nevertheless, MK-801 exposure evoked a distinct movement response (*p*_MK-801_ = 0.043), although no interaction effect was observed (*p*_interaction_ = 0.519). Values are shown as mean ± SEM. * *p* < 0.05.

## Data Availability

The data presented in this study are available as supplementary material containing statistical tables in Appendix A.

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
