# Peer review of "Evidence for Prepulse Inhibition of Visually Evoked Motor Response in Drosophila melanogaster"

_biology, 2023, doi:10.3390/biology12040635_

Round 1
Reviewer 1 Report
The effect of prepulse stimulus on activity related to startle response in flies is demonstrated using a combination of different light intensity, pattern, and with and without a chemical agonist. The authors present a well-written paper and the conclusions are supported by the analyses and discussion. My primary comments are with clarity of terminology and reporting of the interactions of models. The authors should also reference all figures and tables throughout the methods and results where relevant. This is done to some extent, but the tables aren’t referenced at all throughout the paper.
Specific comments:
Simple summary/abstract:
· PPI is introduced as a behavior and then as a method. I appreciate that it could be considered both, but clarifying here and throughout and maintaining consistent terms would be helpful.
Introduction:
· Lines 67-69, I suggest reversing the order of this sentence: one report of ppi in larvae; however, no reports in adult flies...
· DISCO is an excellent name, by the way.
· Line 101, for the naïve reader, more information on what NMDA is, why it’s relevant, and why/how MK-801 works would be helpful.
Materials and methods:
· It isn’t clear from the methods what sexes were used and for what assays. The reader has to work backwards from the results to figure this out. I suggest being clear from the methods what sexes were used and when.
· Line 125, is MK-801 effect dose-dependent in other studies? Why were multiple doses tested?
· Line 126, higher concentrations than 0.6, or because 0.6 was highest this concentration was used?
· Line 141, I appreciate the review of the protocols in the result section, and I think these should be retained. I don’t think that the results section is the only place these details should be found however.
· Consider making your custom protocols in MATLAB available.
· Lines 150-154, this section is very clear. I would only add a line to orient readers relative to plots. Something like positive values indicate increased activity, negative values indicate decreased activity, 0 values are not different from control.
· Lines 156-160, this section suggests that the same 75 flies were tested 8 times. The figure legend makes it sound like a different set of 75 flies were tested at each of the 8 trials. I think it is the former case, but here and in the legend this could be clearer.
· Lines 166-171, the corresponding figure with the assay design isn’t referenced.
· Line 173, “if the movement response was above chance” is it meant above baseline? Above control?
Results:
· Figure 1. Sex isn’t indicated. Are these males and females? See comment above about 75 same flies or different flies for the 8 trials.
· Line 208-9, the lights-off stimuli is part of every assay. Consider referring to it here as startle response. Or, change the heading in figure 1b to startle response. There are small changes/inconsistencies in terminology that make it confusing to keep the assays straight.
· Lines 208-220, the corresponding figure and table isn’t referenced. Also, the interaction isn’t discussed
· Figure 2. A this looks like the same data as presented in 1c. If it is, I suggest analyzing this data once and indicating that the sex model term isn’t significant. In this and other plots, consider using 95% CI instead of SE to assist with visual interpretation. In the legend, model terms are unclear. Is group sex? Is it age? I suggest being specific with model terms in the reporting of p value statistics.
· Line 236, the partial-like pulse presentation is changed from the methods. Reporting in ms is more complex and maybe unnecessary. I suggest using seconds throughout and in figures.
· Line 237-241, instead of plotting the same data different ways, consider showing the main effect of the chemical treatment in 3b and the interaction in figure 3c.
· Lines 246-260, this section doesn’t reference the figure and should after statistics. The table is not referenced. In addition, the authors describe what sounds like an interaction but the interaction term isn’t presented. This (and statistics results) should be added.
· Figure 3. Colors in plots are hard to distinguish in greyscale. Was the control not assessed following exposure to the different concentrations of the chemical? If not, this needs a justification. It seems odd to compare the chemical-treated flies to only non-chemical treated flies.
· Line 277-283, because part of the question focuses on the habituation and part on the exposure to MK-801, it isn’t easily clear which effect is modeled with the term “treatment”. This is another place where terms like “treatment” and “groups” could refer to multiple things. I suggest using “MK-801 exposure” and “trial” for clarity instead.
· Figure 4, in A, it isn’t clear in the diagram when the 1min blinking light exposure occurs. In the legend, “by repeatedly presenting the lights-off stimulus and reducing the interval between trials of lights-off stimuli presentation” could be interpreted as reducing the interval from 10min with each trial. Since there are 9 trials, this could mean that the difference for trial 1 was 10 minutes, the difference between trial 2 was 9 minutes and so on. To avoid this confusion, be explicit what is meant by “reducing the interval between trials”. In B, I suggest making the colors more easily discernable in greyscale.
Discussion:
· Throughout this section reference relevant figures.
· Depending on how you decide to present the 3s 5s and 10s intervals, consider altering the examples referenced in lines 326-342 to match for consistency.
· Line 355, “chance level” is this referring to the control baseline? If it were chance, I would expect that the chance in activity would be as frequently up, down, and no change, but this isn’t consistent with how the calculation of the movement response variable was described.
Author Response
The effect of prepulse stimulus on activity related to startle response in flies is demonstrated using a combination of different light intensity, pattern, and with and without a chemical agonist. The authors present a well-written paper and the conclusions are supported by the analyses and discussion. My primary comments are with clarity of terminology and reporting of the interactions of models. The authors should also reference all figures and tables throughout the methods and results where relevant. This is done to some extent, but the tables aren't referenced at all throughout the paper.
Specific comments:
Simple summary/abstract:
- PPI is introduced as a behavior and then as a method. I appreciate that it could be considered both, but clarifying here and throughout and maintaining consistent terms would be helpful.
We appreciate the suggestion to clarify and maintain consistency in our use of terminology. We have modified the abstract as follows:
- "PPI and habituation learning are widely used tasks" to "PPI is a widely investigated behaviors."
- "PPI is a well-established method that has been observed across various vertebrate and invertebrate species" to "PPI has been observed across various vertebrate and invertebrate species."
Please also note that, following recommendations of Reviewer 2, we refrained from using the term “habituation” in the abstract and manuscript text, as our current data does not fully confirm that the observed reduction of the startle response is habituation learning. We now refer to these results as “reduction of motor response” or “stimuli desensitization”. We hope our changes were able to improve the terminology through our study.
Introduction:
- Lines 67-69, I suggest reversing the order of this sentence: one report of ppi in larvae; however, no reports in adult flies...
We thank the Reviewer for the suggestion. We have revised the sentence as follows: "One report describes acoustic PPI in Drosophila larvae [22]; however, to the best of our knowledge, no studies have been performed demonstrating this behavior in adult flies."
- DISCO is an excellent name, by the way.
We are grateful for the thoughtful comment. We appreciate your positive feedback on the name DISCO.
- Line 101, for the naïve reader, more information on what NMDA is, why it's relevant, and why/how MK-801 works would be helpful.
The Reviewer has a valid concern. We have revised the Introduction section of the manuscript (lines 103-108) to include a brief explanation of the importance of MK-801, as follows:
"Notably, the N-methyl D-aspartate (NMDA) receptor antagonist MK-801 was able to influence Drosophila behavior in both paradigms. MK-801 is commonly used in pre-clinical studies to model phenotypes of schizophrenia in animals, such as sensorimotor gating deficits [12] and, importantly, impaired pre-pulse inhibition [32]. Thus, our results demonstrate the potential use of Drosophila as a model organism for further studies on PPI behavior and its modulation in sensory and motor systems."
Materials and methods:
- It isn't clear from the methods what sexes were used and for what assays. The reader has to work backwards from the results to figure this out. I suggest being clear from the methods what sexes were used and when.
We thank the Reviewer for the helpful comment. We have updated the Methods section (line 117) to include the following sentence: “Unless stated otherwise, all experiments included both male and female animals.” Additionally, we have updated the figures' legends to provide further clarity. We believe this change will improve the accessibility and transparency of our experimental methods.
- Line 125, is MK-801 effect dose-dependent in other studies? Why were multiple doses tested?
In a previous report [Moulin et al. BMC Biol 20, 283 (2022)], we tested the behavioral effects of MK-801 in Drosophila using the same concentrations and found a dose-dependent effect on general activity. Based on these results, we chose to also test multiple doses in this study to explore the potential dose-dependent effects on startle response modulation. The Methods section (lines 127-132) now describes our reasoning as follows:
“Prepulse inhibition experiments were performed by treating the flies with MK-801 (0.6, 0.3, or 0.15 mM) for 24 hours via homogenization in the fly food, a previously described method for drug administration [33]. The chosen concentrations were based on a preceding study from our group [34], where MK-801 effectively replicated hyperlocomotion phenotypes similar to rodent models in a dose-dependent manner.”
We also briefly mention our reference in the Discussion section (lines 420-422). We hope our response clarifies the rationale for using multiple doses in the experiment.
- Line 126, higher concentrations than 0.6, or because 0.6 was highest this concentration was used?
We acknowledge the potential for confusion in our description of the concentration of MK-801 used in the stimuli desensitization protocol. We have revised the manuscript to provide a clearer explanation (lines 132-134), as follows:
“For the stimuli desensitization protocol, we administered the highest concentration used in this study (0.6 mM), as higher doses of MK-801 have been shown to have a greater effect on startle habituation in rodents [35].”
- Line 141, I appreciate the review of the protocols in the result section, and I think these should be retained. I don't think that the results section is the only place these details should be found however.
We agree with the suggestion that the protocol details should not be limited to the results section alone. Thus, we have reviewed and improved the protocol descriptions throughout the manuscript to ensure that the details are clearly stated in all relevant sections.
- Consider making your custom protocols in MATLAB available.
We appreciate the suggestion to make our custom protocols in MATLAB available. We have published similar codes for the control and analysis of DISCO protocols in a previous paper describing the platform [Moulin et al. BMC Biol 20, 283 (2022)], so we have added in the Methods section the information that similar codes can be found in the original article. We hope that this additional information will help interested researchers to find and utilize the codes.
- Lines 150-154, this section is very clear. I would only add a line to orient readers relative to plots. Something like positive values indicate increased activity, negative values indicate decreased activity, 0 values are not different from control.
We thank the Reviewer for the feedback. We agree that a more intuitive explanation of our results and plots was necessary and have revised the manuscript to include a brief sentence regarding the interpretation of the values reflecting changes in activity, now called “startle index” (lines 161-166). The sentence now reads: “In simpler terms, positive index values denote an increased motor response, negative index values indicate a diminished response, and null values signify no deviation from the baseline response”. Moreover, we have added a new panel to figure 1 (as Figure 1C), illustrating in more detail how the index is calculated.
- Lines 156-160, this section suggests that the same 75 flies were tested 8 times. The figure legend makes it sound like a different set of 75 flies were tested at each of the 8 trials. I think it is the former case, but here and in the legend this could be clearer.
We agree that our choice of wording may confuse the readers. To clarify, the same 74 flies were tested eight times, with a rest period between each trial. We understand that the wording could have been more precise and revised accordingly. Additionally, we apologize for a typo in the figure legend, in which we erroneously referred to 75 flies instead of 74. We hope this explanation clarifies the description and we are grateful for the helpful suggestion.
- Lines 166-171, the corresponding figure with the assay design isn't referenced.
We appreciate the helpful observation. We have added a reference to the appropriate figure.
- Line 173, "if the movement response was above chance" is it meant above baseline? Above control?
We apologize for the unclear wording of the manuscript. We revised the sentence to clarify that the movement response was compared to the no-response baseline, not chance or control (lines 197-199). The sentence now reads as follows: “For experiments where we investigated if the movement response was above baseline (i.e., startle index = 0)”
Results:
- Figure 1. Sex isn't indicated. Are these males and females? See comment above about 75 same flies or different flies for the 8 trials.
We thank the Reviewer for the comment. We acknowledge that the sex of the flies in Figure 1 was not indicated, and we apologize for this oversight. To address this issue, we have revised the legend of Figure 1 to indicate that both male and female flies were used in the experiment. Additionally, as mentioned earlier, we have clarified in the Methods section that all experiments, unless stated otherwise, included male and female animals.
- Line 208-9, the lights-off stimuli is part of every assay. Consider referring to it here as startle response. Or, change the heading in figure 1b to startle response. There are small changes/inconsistencies in terminology that make it confusing to keep the assays straight.
We appreciate the insightful suggestion and agree that consistent terminology is important for clarity. As described in the previous comments, we have revised the manuscript to refer to the calculated movement response as "startle index". We have also included an illustration and a more detailed explanation of the index as Figure 1B and the corresponding legend. We believe that these changes will enhance the clarity of our results and make it easier for readers to understand the experimental procedures. Thank you for the helpful feedback.
- Lines 208-220, the corresponding figure and table isn't referenced. Also, the interaction isn't discussed
We are grateful that the Reviewer brought this to our attention. We have made the necessary revisions to the mentioned part of the Results section to include references to the appropriate figure, as well as additional description of the interaction effects (lines 250-270). Additionally, we now describe both in the Methods section and figures how we provided comprehensive tables of all statistical analyses within the Supplementary Data.
- Figure 2. A this looks like the same data as presented in 1c. If it is, I suggest analyzing this data once and indicating that the sex model term isn't significant. In this and other plots, consider using 95% CI instead of SE to assist with visual interpretation. In the legend, model terms are unclear. Is group sex? Is it age? I suggest being specific with model terms in the reporting of p value statistics.
We thank the Reviewer for the valuable feedback. We appreciate the suggestion and agree that presenting the non-independent data in multiple figures should be clearly stated for the readers. To address this issue, we have revised the manuscript to explain that Figures 1C and 2A present data from the same group of flies (lines 250-258). However, we believe that the comparison between males and females is an important aspect of our study, and thus we have kept Figure 2A as a separate figure. We have also made it clear that the "older-adult" group of flies used in Figure 2C is the same as for Figure 1C, while the “younger-adult” flies are a new independent group (lines 263-268).
Regarding the use of 95% CI, we understand the potential benefits of using confidence intervals for interpretation, we believe some confusion may arise from using different statistical measures from what is familiar in the literature. Nevertheless, while we have decided to keep the use of SE in the main figures for consistency with common practice, we have included statistical tables as supplementary data that present the 95% CI for the mean startle index values. We hope that this will provide interested readers with additional information on the precision of our estimates.
Furthermore, we agree that it is important to be specific in our reporting of model terms, and we have revised the legend of Figure 2 to clarify references to the different experimental groups.
- Line 236, the partial-like pulse presentation is changed from the methods. Reporting in ms is more complex and maybe unnecessary. I suggest using seconds throughout and in figures.
We appreciate the Reviewer’s suggestion to clarify the pulse presentation protocol. In summary, the entire infrared monitor is illuminated with 3 parallel LED stripes. It is done in a way that each individual tube containing one fly is illuminated directly by 3 LED lamps, one from each stripe. Thus, turning off one of the stripes will cause, for a given infrared monitor, turning off one of 3 lights that directly illuminate each fly, as illustrated in Figure 1A. In the new version of the manuscript, we improved the description of the dim-light stimuli used as a prepulse (lines 287-292). We hope we have made our explanations clearer now and addressed the Reviewer’s concern.
Moreover, we agree with the reviewer that reporting the prepulse interval in seconds would be more intuitive. Thus, we have changed the report in the text and figures. We are grateful for this insightful suggestion.
- Line 237-241, instead of plotting the same data different ways, consider showing the main effect of the chemical treatment in 3b and the interaction in figure 3c.
We are grateful for the suggestions for the data presentation in Figures 3B and 3C. After these recommendations, we have adapted the existing panels to better illustrate our results as follows:
- In Figure 3B, we now show the main effect of the treatment (MK-801) to highlight its impact on the observed behavior, emphasizing the effects of MK-801 within separate prepulse intervals.
- In Figure 3C, we present the overall effects of trials and the interaction between MK-801 and different prepulse intervals, providing a more comprehensive view of the data.
- Lines 246-260, this section doesn't reference the figure and should after statistics. The table is not referenced. In addition, the authors describe what sounds like an interaction but the interaction term isn't presented. This (and statistics results) should be added.
We appreciate the Reviewer’s feedback. We have now appropriately referenced the figures and statistics results in the mentioned section (now lines 300-328) and have presented the results from statistical analyses more clearly. Additionally, we have cited the statistical tables contained in the Supplementary Data files for readers' reference.
- Figure 3. Colors in plots are hard to distinguish in greyscale. Was the control not assessed following exposure to the different concentrations of the chemical? If not, this needs a justification. It seems odd to compare the chemical-treated flies to only non-chemical treated flies.
We are grateful for the valuable feedback and appreciate the concern about the colors in the plots and the comparison between chemical-treated flies and non-chemical treated flies. To address this comment, we have revised Figure 3 to improve its clarity and ease of interpretation in greyscale. Furthermore, we have added results as Supplementary Figure 4, which show that the startle baselines for the MK-treated flies and the controls are the same in response to the stimuli without a prepulse. Since we observed no significant differences in the startle baselines between the highest dose of our study (0.6mM) and the controls, we decided to focus our efforts on testing the different prepulse times. Based on these results, we believe it is reasonable to assume a similar baseline across groups, allowing for a meaningful comparison of the prepulse inhibition effects. We now describe our rationale in more detail in the Methods section (lines 184-187).
- Line 277-283, because part of the question focuses on the habituation and part on the exposure to MK-801, it isn't easily clear which effect is modeled with the term "treatment". This is another place where terms like "treatment" and "groups" could refer to multiple things. I suggest using "MK-801 exposure" and "trial" for clarity instead.
We thank the Reviewer for pointing out the potential confusion regarding the use of "treatment" and "groups" in lines 277-283. We have revised the text in the Results section referring to Figures 3 and 4, replacing "treatment" with "MK-801 exposure" and "groups" with "trial" to ensure that the terms are clear and unambiguous. We believe that this change will help readers better understand the effects modelled in our study.
- Figure 4, in A, it isn't clear in the diagram when the 1min blinking light exposure occurs. In the legend, "by repeatedly presenting the lights-off stimulus and reducing the interval between trials of lights-off stimuli presentation" could be interpreted as reducing the interval from 10min with each trial. Since there are 9 trials, this could mean that the difference for trial 1 was 10 minutes, the difference between trial 2 was 9 minutes and so on. To avoid this confusion, be explicit what is meant by "reducing the interval between trials". In B, I suggest making the colors more easily discernable in greyscale.
We appreciate the suggestions to improve the clarity and accessibility of the figure. Thus, we have made the following changes: 1. We have created a new version of Figure 1A to better illustrate the 1-minute blinking light exposure protocol. This should provide a clearer representation of the experimental design. 2. We have revised the legend to explicitly describe the process of reducing the interval between trials of lights-off stimuli presentation, avoiding potential misinterpretation. 3. In Figure 4B, we have modified the colors to be more easily discernible in greyscale, enhancing the figure's readability. We hope that these revisions address the concerns and improve the overall quality of our manuscript.
Discussion:
- Throughout this section reference relevant figures.
We are grateful for the Reviewer's suggestions. While we acknowledge that referencing figures in the Discussion section could potentially provide more clarity, we believe that our current writing style maintains the flow and overall coherence of the text. We understand that different authors and reviewers may have varying preferences when it comes to referencing figures within the Discussion; however, in an effort to present a clear and concise narrative, we have chosen not to include figure references within this section. We appreciate the Reviewer’s understanding and hope that our decision does not detract from the overall quality and clarity of the manuscript.
- Depending on how you decide to present the 3s 5s and 10s intervals, consider altering the examples referenced in lines 326-342 to match for consistency.
We appreciate the Reviewer’s attention to detail in pointing out potential inconsistencies regarding the description of the prepulse intervals. We have revised the mentioned lines to ensure consistency with the presentation of the intervals throughout the manuscript.
- Line 355, "chance level" is this referring to the control baseline? If it were chance, I would expect that the chance in activity would be as frequently up, down, and no change, but this isn't consistent with how the calculation of the movement response variable was described.
We thank the Reviewer for drawing our attention to the potential confusion surrounding the term "chance level" in line 355. To address this issue, we have replaced "chance level" with a more adequate description, as the following (lines 460-462):
“After four trials, the startle index reached null levels, indicating that the movement response was inhibited. These results suggest that the repeated presentation of the lights-off stimuli was capable of a response reduction.”
We believe this change should better align with the calculation of the startle index and provide a clearer understanding of our results. We appreciate the valuable comments and suggestions and are grateful for the help in improving our manuscript.
Reviewer 2 Report
In their manuscript, Schioth, et al. describe new behavioral assays using their new DISCO apparatus. The assays include prepulse inhibition (PPI) and habituation of motor responses to lights-off stimuli. The assays are mostly well described, and I believe will be useful additions to the Drosophila behavioral toolbox. However, the current description is terse, with room for increased clarity. Moreover, the experimental tests of the behaviors are incomplete, and there are questions regarding the statistical tests.
The primary behavioral readout for PPI is a movement response to lights-off. It is not clear how this measure was calculated exactly. The authors refer back to their 2022 paper, but a reader of this paper should be able to readily assess the central variable without seeking other papers.
The authors used t-tests and ANOVA to assess significant differences in the groups used. However, they never showed that the data were normally distributed; hence, these tests may be inappropriate. The authors should either demonstrate normality or use different tests. Even if the data are normal, I would like the authors to justify using the Holm-Sidak posthoc test. While this test can offer more power it fails to measure confidence intervals. Moreover, please report for each ANOVA, the F score with degrees of freedom, the N, and the probability.
Fig1C demonstrates a significant movement response after lights-off over 8 trials. The average of all eight trials is used to demonstrate the significant effect. Is there a significant effect of trial number on movement response? If so, they probably should not be averaged for this t-test. How did they decide on eight trials?
Figure 2 is not reliably cited in the results text, nor are the statistics. Fig 2A shows the same dip in trial 3 as shown in Fig 1C, which could be interesting if these are independent experiments. How consistently is this dip found, and if consistent, would it be trial-dependent or time-dependent (e.g., ISI or ITI)? If the experiments are not independent, that is also critical to report as it affects one's judgment about the reproducibility of the assay.
In the experiment shown in Fig 2B, testing the effect of female reproductive status on movement response, is there a significant increase in movement response, as was seen in Fig 1C? If a movement response is not seen, one cannot expect to detect a difference between groups.
In Figure 3, the authors test the effect of a dimmed prepulse on the movement response to a full lights-off. The prepulse was given at 3, 5, and 10 seconds before the full lights-off stimuli and averaged over 8 trials. The authors also test for criteria validity of PPI, but using the NMDA inhibitor MK-801. I have several concerns with this experiment.
My first concern is, again, whether the trial number affects the movement response, with or without the prepulse. All the data are collapsed into a single mean for each time point. A second concern/question is whether. MK-801 has an effect on movement response without the PP. Since the movement effect seems to return with a 10 second ISI, there should be a normal response, but the authors have not shown us this. A third concern is the use of the reported MK-801 dosages and their possible interaction with PP timing. It would be best to have a low dose below the response threshold. Although 0.15mM is not significantly different across all PP intervals, it could be at the 3-second interval. Using a different statistical test may reveal this interaction.
Since MK-801 is an NMDA antagonist that produces many negative effects of schizophrenia in rodents, its application here would appear to strive towards reaching a criterion validity for PPI. However, NMDA receptors in rodents and Drosophila are likely to have different roles, given the difference in glutamate signaling between the two species. Nevertheless, demonstrating a role for NMDA receptors would significantly contribute to understanding the mechanisms underlying PPI. This demonstration would rely on using more than one antagonist. I recommend using either an NMDA subunit mutant or an induced RNAi transgene to map the defects back to the receptor.
Minor points related to Figure 3, the PPI interval can be more simply reported as seconds rather than 10^3 msec. Also, since the Y-axis in panel B does not align with panel C, the Fig 3C Y-axis should be labeled independently.
In Figure 4, the authors attempt to demonstrate the habituation of the light response through a relatively blinking light presentation. However, they failed to demonstrate that the diminishing response was habituation (does it dishabituate, is there an effect of stimulus intensities, does it recover spontaneously, etc...?; please see: https://doi.org/10.1016/j.neubiorev.2017.05.028). Without address at least some of these criteria, this response decrement cannot be claimed to be habituation. Moreover, the role of NMDA channels in this response decrement is also missing here.
Author Response
In their manuscript, Schioth, et al. describe new behavioral assays using their new DISCO apparatus. The assays include prepulse inhibition (PPI) and habituation of motor responses to lights-off stimuli. The assays are mostly well described, and I believe will be useful additions to the Drosophila behavioral toolbox. However, the current description is terse, with room for increased clarity. Moreover, the experimental tests of the behaviors are incomplete, and there are questions regarding the statistical tests.
The primary behavioral readout for PPI is a movement response to lights-off. It is not clear how this measure was calculated exactly. The authors refer back to their 2022 paper, but a reader of this paper should be able to readily assess the central variable without seeking other papers.
We appreciate the feedback on the clarity of our manuscript and the need for a more detailed explanation of our movement response measure. We have added a new panel to Figure 1 (now Figure 1C), which provides a clear illustration of how our movement response measure is calculated. We hope it can help readers better understand the central variable without needing to refer to our previous publication. Additionally, in line with the suggestions from Reviewer 1, we have changed the term "movement response" to "startle index" throughout the manuscript. We believe this new term improves clarity and better reflects the nature of the measurement.
The authors used t-tests and ANOVA to assess significant differences in the groups used. However, they never showed that the data were normally distributed; hence, these tests may be inappropriate. The authors should either demonstrate normality or use different tests. Even if the data are normal, I would like the authors to justify using the Holm-Sidak posthoc test. While this test can offer more power it fails to measure confidence intervals. Moreover, please report for each ANOVA, the F score with degrees of freedom, the N, and the probability.
Thank you for your suggestion. In response, we have added a dot-plot panel to Figure 1 (Fig 1E) to display single-fly responses across trials and the mean response for all trials. While we recognize that individual behavior varies greatly, we would like to emphasize that the group effect is indeed robust (p=0.0004, n=74 male and female flies, one-sample t-test against 0). Furthermore, the histogram of mean startle values indicates no significant departure from normal distribution (p=0.052, D'Agostino & Pearson normality test). Moreover, the choice of D'Agostino & Pearson normality test was made thoughtfully, as it is not only recommended by GraphPad statisticians but also supported by various sources for its versatility and power [10.1093/biomet/60.3.613]. Given that our primary experimental measure with the largest sample size displayed a normal distribution, we found it sensible to assume normality for subsequent tests, as widely advised in the literature [10.1186/1471-2288-12-81; 10.1080/00949657808810243].
Additionally, the Holm-Sidak method is similar to the Bonferroni process, but with more power [10.2105/ajph.86.5.726]. The main distinction is that by the Holm-Sidak calculation, the confidence interval of the difference between means cannot be computed. However, the p-value and the confidence interval of the difference are intended to provide two complementary pieces of information: the p-values assess null hypothesis acceptance and highlight significant findings, which is especially valuable for exploratory studies, while the confidence intervals indicate the range of the effect’s true values, direction, and strength, allowing inference of the results’ relevance [10.3238/arztebl.2009.0335].
Thus, while we acknowledge that it does not provide confidence intervals adjusted for multiple comparisons, our primary focus was on confidently determining whether PPI occurred and if it could be reverted by any extent after MK-801 treatment, for which the p-values provided by the Holm-Sidak method are sufficient. Considering all the listed advantages of the Holm-Sidak test, we have chosen to keep this methodology. Nevertheless, to accommodate any specific interest from readers in confidence intervals, we have calculated the unadjusted 95% confidence intervals of the differences, which can be found within the statistical tables in the Supplementary Data. We have also included a detailed explanation of our rationale in choosing the D'Agostino & Pearson normality test and the Holm-Sidak method in the Methods section (lines 195-214).
We hope we have addressed the Reviewer concerns and have provided a clear justification for our choice of statistical tests. We appreciate the valuable input and the opportunity to improve the rigor and clarity of our manuscript. If the Reviewer has further interest in the Holm-Sidak method, the calculations used by the analysis software can be found at (Maxwell, S. E., Delaney, H. D., & Kelley, K. (2017). Designing experiments and analyzing data: A model comparison perspective. Routledge.)
Fig1C demonstrates a significant movement response after lights-off over 8 trials. The average of all eight trials is used to demonstrate the significant effect. Is there a significant effect of trial number on movement response? If so, they probably should not be averaged for this t-test. How did they decide on eight trials?
The question if there is a significant effect of trial number is inadequate to Figure 1C (currently Figure 1D) since it involves a one-group sample, which precludes testing for systematic trial effects. Although trial 3 deviates in this sample, it doesn't establish an overall trial effect. Moreover, no such effect is observed in the independently sampled groups in Figure 2, except for the male vs. female comparison originating from Fig 1C's sample, as now described in the Results section (lines 255-257). Consequently, averaging the trials for the t-test remains a valid approach. Furthermore, the Methods section now explains the decision to use eight trials, taking into account the flies' susceptibility to dehydration after several hours of experiment (lines 169-176).
Figure 2 is not reliably cited in the results text, nor are the statistics. Fig 2A shows the same dip in trial 3 as shown in Fig 1C, which could be interesting if these are independent experiments. How consistently is this dip found, and if consistent, would it be trial-dependent or time-dependent (e.g., ISI or ITI)? If the experiments are not independent, that is also critical to report as it affects one's judgment about the reproducibility of the assay.
We appreciate the attention to the details of our manuscript and the opportunity to clarify the concerns raised regarding the results. We agree that presenting non-independent data in multiple figures should be clearly stated for the readers. To address this issue, we have revised the manuscript to explain that Figures 1C and 2A present data from the same group of flies (lines 251-254) and we have ensured that all the experiments are appropriately described as independent or non-independent, as applicable, to provide a clear understanding of the experimental design and the reliability of our findings. We also have revised the Results section to better describe the samples and to include the relevant statistics.
Regarding the dip observed in trial 3 in both Figures 1C and 2A, we would like to reiterate that our experiments show no systematic effect of a specific trial. Moreover, it is important to note that in our previous study using a similar protocol to assess the same behavior, we found no significant effect of trial number when testing three different strains of flies across six trials conducted over three hours. What we observed, consistent with the findings in the current study, is that this behavior is highly variable, and individual trials can elicit different results. We believe that the average of the trials is the most important factor to consider in this context.
We hope that our revised Results section and this response address your concerns and provide the necessary clarity regarding these results. We are grateful for the valuable input in helping us improve our work.
In the experiment shown in Fig 2B, testing the effect of female reproductive status on movement response, is there a significant increase in movement response, as was seen in Fig 1C? If a movement response is not seen, one cannot expect to detect a difference between groups.
We believe the Reviewer raises a valid concern. Thus, we have restructured the data presentation in all panels of Figure 2 to ensure that there is a significant increase in movement response. Similar to Figure 1, we have now calculated and displayed the averages of the trials and performed one-sample t-tests for all tested groups. Our analysis confirmed that there was indeed a significant increase in movement response for the female flies, allowing us to detect differences between the groups. This information has also been incorporated into the manuscript text and figure legend to provide further clarity on the results.
In Figure 3, the authors test the effect of a dimmed prepulse on the movement response to a full lights-off. The prepulse was given at 3, 5, and 10 seconds before the full lights-off stimuli and averaged over 8 trials. The authors also test for criteria validity of PPI, but using the NMDA inhibitor MK-801. I have several concerns with this experiment.
My first concern is, again, whether the trial number affects the movement response, with or without the prepulse. All the data are collapsed into a single mean for each time point. A second concern/question is whether. MK-801 has an effect on movement response without the PP. Since the movement effect seems to return with a 10 second ISI, there should be a normal response, but the authors have not shown us this. A third concern is the use of the reported MK-801 dosages and their possible interaction with PP timing. It would be best to have a low dose below the response threshold. Although 0.15mM is not significantly different across all PP intervals, it could be at the 3-second interval. Using a different statistical test may reveal this interaction.
We thank the Reviewer for the relevant concern. Indeed, Figure 3 shows only the final averages of independent 8-trial experiments, and we agree that it would be useful for the reader’s interpretation to have access to these results. Thus, Supplementary Figures 1-3 have been added to display the trial-by-trial response for each MK-801 dose and prepulse interval (PPI), along with the corresponding statistics in the figure legends. As mentioned in previous comments, despite some trials showing significant effects in the ANOVA analyses, there is no systematic effect attributable to a specific trial or the passage of time. We ascribe this effect to the highly variable nature of this behavior, that again, has been demonstrated in prior research. Moreover, it is not uncommon for Drosophila behavioral experiments to be evaluated in terms of population-level effects and through several-point or continuous measurements that are later averaged. This approach is designed to assess consistent population-level behavioral patterns, taking into account the high individual variability inherent in this model organism.
Moreover, we have incorporated Supplementary Figure 4, which demonstrates that the startle baselines for flies exposed to MK-801 at the highest dose in our study (0.6mM) and controls are equivalent in response to stimuli without a prepulse. Given that we observed no significant differences in startle baselines between the MK-801-treated group and the controls, we opted to concentrate our efforts on examining various prepulse intervals. Based on these findings, we consider it reasonable to assume a consistent baseline across groups, thus enabling a meaningful comparison of the prepulse inhibition effects. We have now elaborated on our rationale in greater detail within the Methods section (lines 184-187).
Lastly, we agree with the Reviewer that a more detailed investigation of how different MK-801 concentrations affect various prepulse intervals (PPIs) would indeed enhance data interpretation. As described in the Results section and Supplementary Data, we have now conducted one-way ANOVAs within each PPI assessment. This analytical approach was selected because experiments involving different PPI times are entirely independent, but comparisons within the same PPI require multiple comparison corrections, as same-PPI controls were grouped together. The Methods section (lines XX-XX) now outlines these analytical steps in a similar manner. Our findings indicate that the only significant within-PPI effects occurred at the 3 sec-PPI, with exposure to the 0.3mM and 0.15mM concentrations, as the reviewer estimated. Consequently, we acknowledge that while the 0.15mM concentration is not significantly different across all PP intervals, it would be ideal to have a concentration that exhibits no effects throughout all experiments. We now address this limitation of our study in the Discussion (lines 440-447) section as follows:
“We should keep in mind that although the 0.6 mM concentration had no significant influence throughout experiments, it would be ideal to also observe a smaller concentration with no effects to serve as a valuable baseline for comparison. While our findings provide valuable insights into the role of NMDA receptors in prepulse inhibition, we recognize they also raise further questions on their mechanistic details and suggest future studies to include a wider range of MK-801 concentrations to establish a more refined dose-response relationship.”
Since MK-801 is an NMDA antagonist that produces many negative effects of schizophrenia in rodents, its application here would appear to strive towards reaching a criterion validity for PPI. However, NMDA receptors in rodents and Drosophila are likely to have different roles, given the difference in glutamate signaling between the two species. Nevertheless, demonstrating a role for NMDA receptors would significantly contribute to understanding the mechanisms underlying PPI. This demonstration would rely on using more than one antagonist. I recommend using either an NMDA subunit mutant or an induced RNAi transgene to map the defects back to the receptor.
We appreciate the suggestion to explore the role of NMDA receptors in both Drosophila and rodents. Thus, we have added a comprehensive discussion about the roles of NMDA across species in the Discussion section (lines 416-447). There, we highlight the conserved homologues of key mammalian genes encoding NMDA receptor subunits in Drosophila and the similarities in NMDA receptor signaling and neuronal plasticity across species. We also address the pharmacological parallels between MK-801 interactions with mammalian and invertebrate NMDA receptors. It reads as follows:
“We tested the effects of MK-801 exposure on PPI. This compound is a selective NMDA receptor antagonist widely used in preclinical research and has been previously employed disruption of PPI in rodent models [32]. Notably, the binding site of MK-801-like drugs has been shown as essential for PPI-inhibition effects [46]. We observed that the prepulse effect was partially disrupted by exposing the flies to MK-801. We employed three different concentrations based on previous experiments describing the effects of MK-801 on Drosophila behaviour [34]. First, 0.15 mM MK-801 partially disrupted PPI within the 3-second interval testing, although we did not detect significant effects across all other intervals. The 0.3 mM group also exhibited a significant PPI reversion for the 3-second interval and was the only exposure that exhibited a significant effect when considering all prepulse intervals. Conversely, MK-801 at 0.6 mM did not influence PPI in any of our analyses. This pattern of effect magnitudes across tested concentrations may be a noteworthy finding, as it might suggest an inverted U-shaped dose-response curve between NMDA inhibition and PPI disruption in flies.
Importantly, Drosophila features conserved homologues of key mammalian genes encoding NMDA receptor subunits, specifically NR1 and NR2, which exhibit common genetic and structural attributes [47]. For instance, the NR1 glycine-binding and NR2 glutamate-binding domains display conserved amino acid sequences [48–50], and functional similarities in NMDA receptors concerning signalling and neuronal plasticity are well-established [34,50–52]. Moreover, pharmacological parallels between MK-801 interactions with mammalian and invertebrate NMDA receptors have been identified, including blocking NMDA-dependent processes [52–54] and targeting the conserved asparagine residue in NR1 subunits[49–51,54]. Thus, the observed effects of MK-801 not only reinforce the validity of the results but also highlight the potential of using Drosophila as a model system o further investigate the neural mechanisms underlying PPI. Nevertheless, we should keep in mind that although the 0.6 mM concentration had no significant influence throughout experiments, it would be ideal to also observe a smaller concentration with no effects to serve as a valuable baseline for comparison. While our findings provide valuable insights into the role of NMDA receptors in prepulse inhibition, we recognize they also raise further questions on their mechanistic details and suggest future studies to include a wider range of MK-801 concentrations to establish a more refined dose-response relationship.”
However, we would like to clarify that our study's primary goal was to demonstrate the previously undescribed PPI behavior in adult flies as a proof-of-principle. While we agree that investigating the role of NMDA receptors in PPI using mutants or RNAi transgenes would be an interesting avenue for future research, we believe these investigations are outside the scope of our current study.
Nevertheless, we recognize the importance of supporting upcoming research on NMDA receptor function and have included a brief statement in the Conclusion section of the revised manuscript to encourage other researchers to explore this topic in more detail in the future (lines 509-515). It reads as follows:
"Moreover, our results indicate that NMDA receptors may play similar roles in prepulse inhibition across species. We believe further investigations, potentially integrating genetic tools, could contribute to unravelling the neural mechanisms behind the PPI paradigm. With further refinement and combined with the Drosophila toolbox for manipulating specific genes and neuronal circuits, behavioural tests like those presented here can significantly advance our understanding of psychiatric disorders by enabling drug screenings or genomic studies"
Minor points related to Figure 3, the PPI interval can be more simply reported as seconds rather than 10^3 msec. Also, since the Y-axis in panel B does not align with panel C, the Fig 3C Y-axis should be labeled independently.
We thank the Reviewer for pointing out these issues. We have modified the PPI interval labels in the figure to represent seconds instead of 10^3 milliseconds for simplicity and better readability. We have added an independent Y-axis label for panels B and C to ensure clarity. We hope these changes improve the presentation and interpretation of the figure.
In Figure 4, the authors attempt to demonstrate the habituation of the light response through a relatively blinking light presentation. However, they failed to demonstrate that the diminishing response was habituation (does it dishabituate, is there an effect of stimulus intensities, does it recover spontaneously, etc...?; please see: https://doi.org/10.1016/j.neubiorev.2017.05.028). Without address at least some of these criteria, this response decrement cannot be claimed to be habituation. Moreover, the role of NMDA channels in this response decrement is also missing here.
We appreciate your concern about the demonstration of habituation in Figure 4. We agree that although our results may represent a first step in that direction, our current data does not fully confirm that the observed reduction of the startle response is habituation learning. Thus, we have revised the terminology throughout the manuscript to clarify that we are observing a reduction in startle response, which might be an element of habituation. Specifically, we refrained from using the term “habituation” in the Title and Abstract, and now refer to these results as “reduction of motor response” or “stimuli desensitization”. Moreover, in the Introduction section, we have described the reduction in startle response as one of the elements of habituation. Lastly, in the Discussion, we have emphasized that this reduction in startle response is possibly coming from habituation learning but acknowledged that further investigation is needed to confirm this claim, citing the provided reference.
Reviewer 3 Report
TITLE: “Evidence for prepulse inhibition and habituation of visually evoked motor response in Drosophila melanogaster”
BRIEF SUMMARY:
The authors present a proof-of-principle paper using a low-cost DISCO platform to mimic predator conditions (light blocked by vicinity of predator) to elicit an escape response in adult flies and quantify the behavioral output. In addition, they modify the protocol to introduce a preemptive dimming of light to serve as a weak stimulus known as prepulse inhibition, a phenomenon known to diminish the magnitude of behavioral responses and implicated in neurological conditions such as schizophrenia in humans but had not been previously reported in adult flies. Furthermore, they demonstrate pharmaceutical intervention of the prepulse inhibition by the MK-801 compound, which has also been shown to modify prepulse inhibition in rats. Finally, they demonstrate the utility of the DISCO platform in performing habituation studies in flies in using fly movement as a biological readout. The straightforward setup and relatively low cost of the DISCO platform provides an attractive system for pharmacogenetic intervention at the intersection of neurobiology and behavioral studies.
COMMENTS:
Overall, I find this paper very interesting and of high significance for (at least) three reasons:
1) In terms of basic biology, the experiments provide the first known evidence of prepulse inhibition (PPI) in adult flies, which adds to the growing list of diverse species that exhibit PPI as well, underscoring the conservation of this neurobehavioral process. It also presents significant support for the DISCO platform behavior readout system as a straightforward alternative for quantitative habituation studies in fruit flies.
2) By using the DISCO platform to demonstrate significant changes in PPI with a known pharmaceutical agent (MK-801) that also works in mammalian model organisms such as rats, this provides validation for the use of fruit flies and pharmacogenetic screens for molecules that could potentially treat various neurological conditions in humans.
3) The simplified parameters and low cost in setting up the DISCO platform for quantitative analysis of neurobiology and behavioral studies should enable more researchers with modest budgets to undertake a similar setup with the promise of obtaining robust data for further analysis, furthering reproducibility in the process.
I have some general comments and suggestion mainly on the presentation of the data in the figures (as well as expanding upon descriptions in captions and methods):
1) I very much appreciate that the illustration of the protocols makes it easy to understand how the DISCO platform works and the modifications made to the original startle-response paradigm to introduce prepulse inhibition. The line and bar graphs also make it easy to see the effect of prepulse inhibition on the escape response in flies with and without pharmaceutic intervention (MK-801). However, it is not clear what the movement response values represent on the X-axes in the figures. In Figure 1, the trials of movement response during straightforward darkness stimulation of escape response shows an overall baseline of 0.2-0.4. The methods state that the movement response is (average movement counts before stimulus) - (average movement counts after stimulus). But the X-axis only contains a value range between 0.0 and 0.4 and the graphs imply that the closer you get to 0.0, the more significant the movement decrease is, and the closer to 0.4, the more significant increase there is in movement. Is the true range here 0.0 to 1.0, in which 1.0 is equivalent to crossing the 17 infrared beams? If so, it would make sense then why the movement response value is always less than 1.0 because no fly will cross all 17 (or even the majority of the) beams in 3 seconds. If this is correct, I would strongly recommend adding this information to make it easier to understand what the actual calculation is, because the illustration shows the fly crossing multiple beams with an equivalent movement response of 0.2-0.4, which can be confusing to readers if they are expecting each beam cross to be counted as a full integer.
2) For figures 1 and 2, why is the third trial very close to 0.0 for movement response across the various conditions? From what I understand, each trial has a 30-minute wait in between to prevent habituation, but it seemed like they weren't responding much at all in the third trial.
3) The dotted line at 0.0 in the graphs of Figure 1 and 2 do not have an explanation in the caption or methods. I would propose that the dotted line at 0.0 be labelled as "no movement" and the basal movement response of wildtype flies (between 0.2 and 0.4) as an additional dotted line. I think this would help the reader immensely in understanding the effectiveness of prepulse inhibition in reducing the magnitude of the escape response and how molecules such as MK-801 can interfere with prepulse inhibition.
4) I am curious why the Holms-Sidak post hoc test was used instead of Bonferroni. Do these types of studies avoid computing confidence intervals because of a large natural variance in stimulation-dependent behavioral responses? If so, it might be helpful to include that qualification in the methods, so that readers have a better understanding of the rationale for using this test.
5) It was good to see evidence that sex, age, and reproductive status do not possess a significant effect on the movement response of the flies, which supports the DISCO platform and the fruit fly model organism as straightforward and robust choices for this kind of research. I am curious if the trials were conducted at the same timepoint of the day? Considering that circadian rhythms can influence the physiological function of the giant fiber system, some readers might wonder if the time of day could have a potential influence on the movement response.
6) This is more of a general comment, but in thinking about the effects of prepulse inhibition and habituation on fruit flies living outside in nature, I have to wonder if there might be potential for weather conditions to affect the ability of fruit flies to accurately escape predators in some scenarios. For instance, a partly cloudy day could conceivably alternate the intensity of the sunlight from bright to dim if clouds pass in front of the sun and that the frequency of the changing intensities could be the difference between prepulse inhibition (that favors predator success) and habituation (in which fruit flies determine that repeated changes in intensity of the sunlight can be safely ignored). If so, could weather affect prepulse inhibition and habituation in humans with schizophrenia and other neuroaffective conditions?
7) The manuscript was very easy to read and was also concise, yet detailed. The comments above are all (in my eyes) just minor revisions to make, and I would love to see these additions to the manuscript to make it even more accessible and easier to understand for potential readers.
Author Response
TITLE: "Evidence for prepulse inhibition and habituation of visually evoked motor response in Drosophila melanogaster"
BRIEF SUMMARY:
The authors present a proof-of-principle paper using a low-cost DISCO platform to mimic predator conditions (light blocked by vicinity of predator) to elicit an escape response in adult flies and quantify the behavioral output. In addition, they modify the protocol to introduce a preemptive dimming of light to serve as a weak stimulus known as prepulse inhibition, a phenomenon known to diminish the magnitude of behavioral responses and implicated in neurological conditions such as schizophrenia in humans but had not been previously reported in adult flies. Furthermore, they demonstrate pharmaceutical intervention of the prepulse inhibition by the MK-801 compound, which has also been shown to modify prepulse inhibition in rats. Finally, they demonstrate the utility of the DISCO platform in performing habituation studies in flies in using fly movement as a biological readout. The straightforward setup and relatively low cost of the DISCO platform provides an attractive system for pharmacogenetic intervention at the intersection of neurobiology and behavioral studies.
COMMENTS:
Overall, I find this paper very interesting and of high significance for (at least) three reasons:
1) In terms of basic biology, the experiments provide the first known evidence of prepulse inhibition (PPI) in adult flies, which adds to the growing list of diverse species that exhibit PPI as well, underscoring the conservation of this neurobehavioral process. It also presents significant support for the DISCO platform behavior readout system as a straightforward alternative for quantitative habituation studies in fruit flies.
We are pleased with the positive feedback on our study. We are indeed excited to contribute to the growing understanding of PPI across diverse species and to showcase the potential of the DISCO platform as an alternative for behavioral studies in fruit flies. The recognition of our work’s significance is appreciated, and we hope our findings can inspire further research in this area.
2) By using the DISCO platform to demonstrate significant changes in PPI with a known pharmaceutical agent (MK-801) that also works in mammalian model organisms such as rats, this provides validation for the use of fruit flies and pharmacogenetic screens for molecules that could potentially treat various neurological conditions in humans.
We thank the Reviewer for the comments on the value of our work. We believe that demonstrating the effects of MK-801 on PPI in Drosophila not only supports the validation of this neurobehavioral process across species but also highlights the potential for using Drosophila as an important model for future studies on the molecular basis of neurological conditions. We hope our findings can contribute to the advancement of research in this area.
3) The simplified parameters and low cost in setting up the DISCO platform for quantitative analysis of neurobiology and behavioral studies should enable more researchers with modest budgets to undertake a similar setup with the promise of obtaining robust data for further analysis, furthering reproducibility in the process.
We are grateful for the thoughtful evaluation of the DISCO platform benefits, especially its simplified parameters and low cost. We believe that making such a platform accessible to researchers with modest budgets will indeed facilitate the generation of robust data and contribute to the reproducibility of findings in the field of neurobiology and behavioral studies. We appreciate the kind words on the potential impact of our work and hope that it will encourage more researchers to adopt and explore the capabilities of the DISCO platform.
I have some general comments and suggestion mainly on the presentation of the data in the figures (as well as expanding upon descriptions in captions and methods):
1) I very much appreciate that the illustration of the protocols makes it easy to understand how the DISCO platform works and the modifications made to the original startle-response paradigm to introduce prepulse inhibition. The line and bar graphs also make it easy to see the effect of prepulse inhibition on the escape response in flies with and without pharmaceutic intervention (MK-801). However, it is not clear what the movement response values represent on the X-axes in the figures. In Figure 1, the trials of movement response during straightforward darkness stimulation of escape response shows an overall baseline of 0.2-0.4. The methods state that the movement response is (average movement counts before stimulus) - (average movement counts after stimulus). But the X-axis only contains a value range between 0.0 and 0.4 and the graphs imply that the closer you get to 0.0, the more significant the movement decrease is, and the closer to 0.4, the more significant increase there is in movement. Is the true range here 0.0 to 1.0, in which 1.0 is equivalent to crossing the 17 infrared beams? If so, it would make sense then why the movement response value is always less than 1.0 because no fly will cross all 17 (or even the majority of the) beams in 3 seconds. If this is correct, I would strongly recommend adding this information to make it easier to understand what the actual calculation is, because the illustration shows the fly crossing multiple beams with an equivalent movement response of 0.2-0.4, which can be confusing to readers if they are expecting each beam cross to be counted as a full integer.
We are grateful for the thoughtful feedback on the presentation of our data in the figures. We appreciate the suggestion to clarify the movement response values. We have now included a new panel in Figure 1 and a more detailed explanation in the manuscript text to help readers better understand the movement response calculation, now referred to as “startle index” following recommendations from Reviewer 1.
Briefly, these values are not a direct representative of the number of beams crossed by the fly, instead they represent the change in the number of times the flies crossed any beams (i.e. movement counts). For that reason it is calculated as (average movement counts before stimulus, the baseline) - (average movement counts after stimulus, the response). We have added a note in the figure caption to clarify the calculation of the movement response and its representation in the graph. This additional information should help readers interpret the data more accurately and understand the relationship between movement response and the observed effects on escape response.
2) For figures 1 and 2, why is the third trial very close to 0.0 for movement response across the various conditions? From what I understand, each trial has a 30-minute wait in between to prevent habituation, but it seemed like they weren't responding much at all in the third trial.
The Reviewer raises an important question. The observed movement response values around 0.3 on average indicate one extra movement within the 3-second window that we measured. However, it is important to note that even lower values might be expected, as our previous research has shown that red-eyed flies are generally not responsive in most trials. However, when examining the group response after several trials, a robust difference from zero (i.e. no changes in movement) is observed.
The lack of response in the third trial can be attributed to the inherent variability in fly behavior. While the flies are not responding much in the third trial, the averages of the overall response pattern demonstrate a robust group-level motor reaction to the lights-off stimuli. We hope our explanation could clarify the raised concerns.
3) The dotted line at 0.0 in the graphs of Figure 1 and 2 do not have an explanation in the caption or methods. I would propose that the dotted line at 0.0 be labelled as "no movement" and the basal movement response of wildtype flies (between 0.2 and 0.4) as an additional dotted line. I think this would help the reader immensely in understanding the effectiveness of prepulse inhibition in reducing the magnitude of the escape response and how molecules such as MK-801 can interfere with prepulse inhibition.
We thank the Reviewer for the suggestion to improve the clarity of Figures 1 and 2. We have taken the Reviewer’s advice and added better descriptions in the figure legends. Now we have better described in the figured legends that the dotted line at 0.0 represents no change in movement.
4) I am curious why the Holms-Sidak post hoc test was used instead of Bonferroni. Do these types of studies avoid computing confidence intervals because of a large natural variance in stimulation-dependent behavioral responses? If so, it might be helpful to include that qualification in the methods, so that readers have a better understanding of the rationale for using this test.
We appreciate the Reviewer’s interest in our choice to use the Holm-Sidak post hoc test instead of the Bonferroni test. The Holm-Sidak method is similar to the Bonferroni process, but with more power [10.2105/ajph.86.5.726]. The main difference is that, with the Holm-Sidak calculation, the confidence interval of the difference between means cannot be computed. Statistically, the p-value and the confidence interval of the difference provide two complementary pieces of information: The p-values assess null hypothesis acceptance and highlight significant findings, which is especially valuable for exploratory studies, while the confidence intervals indicate the range of the effect’s true values, direction, and strength, allowing inference of the results’ relevance [10.3238/arztebl.2009.0335].
In our study, the focus was not on confidence interval calculations, as we were primarily interested in describing with confidence whether PPI was found and if it could be reverted to some extent after MK-801 treatment. We have now explained this rationale in the Methods section of the manuscript (lines 195-215). We hope this explanation helps clarify our choice and provides a better understanding of our statistical approach.
5) It was good to see evidence that sex, age, and reproductive status do not possess a significant effect on the movement response of the flies, which supports the DISCO platform and the fruit fly model organism as straightforward and robust choices for this kind of research. I am curious if the trials were conducted at the same timepoint of the day? Considering that circadian rhythms can influence the physiological function of the giant fiber system, some readers might wonder if the time of day could have a potential influence on the movement response.
We thank the Reviewer for the relevant question. In the Methods section, we now added that throughout the study, experiments were conducted during the light phases (ZT 0-12) of the light/dark cycle, encompassing the entire manipulation and testing of flies (lines 122-124). Though not systematically investigated, we observed no apparent differences in responses within this circadian rhythm range. However, potential influences may exist and require closer examination due to the behavior's variability – offering an interesting avenue for future research.
6) This is more of a general comment, but in thinking about the effects of prepulse inhibition and habituation on fruit flies living outside in nature, I have to wonder if there might be potential for weather conditions to affect the ability of fruit flies to accurately escape predators in some scenarios. For instance, a partly cloudy day could conceivably alternate the intensity of the sunlight from bright to dim if clouds pass in front of the sun and that the frequency of the changing intensities could be the difference between prepulse inhibition (that favors predator success) and habituation (in which fruit flies determine that repeated changes in intensity of the sunlight can be safely ignored). If so, could weather affect prepulse inhibition and habituation in humans with schizophrenia and other neuroaffective conditions?
The Reviewer’s comment regarding the potential influence of weather conditions on prepulse inhibition and habituation in both fruit flies and humans is indeed an interesting and thought-provoking perspective. It is plausible that environmental factors, such as changing light conditions on a partly cloudy day, could affect the sensory processing of fruit flies and, in turn, influence their responses to potential threats.
In the context of humans with schizophrenia, it is also possible that weather conditions might have an impact on sensory processing and, subsequently, on PPI and habituation. We believe that may be an interesting topic for further research aiming to determine the extent of these effects and whether they hold significant implications for individuals with these conditions.
We appreciate your insightful comment and have added a brief mention of this idea in the Discussion section of the manuscript, inviting future research to explore this interesting connection further (lines 466-472). It reads as follows:
“Additionally, the successful observation of PPI after using dim light as a prepulse stimulus suggests that it may be worth considering future studies on the potential influence of environmental factors, such as weather or daylight conditions, on prepulse inhibition. For instance, fluctuating light conditions in natural settings could affect sensory processing, potentially impacting the responses to various threatening stimuli. Exploring these connections could provide valuable insights into how environmental light settings may contribute to the sensory and behavioural responses across species.”
7) The manuscript was very easy to read and was also concise, yet detailed. The comments above are all (in my eyes) just minor revisions to make, and I would love to see these additions to the manuscript to make it even more accessible and easier to understand for potential readers.
We are thankful for the positive feedback and helpful suggestions. We much appreciate the thorough review and insights, which have helped improve the clarity and accessibility of our manuscript.